# Knowledge synthesis of 100 million biomedical documents augments the deep expression profiling of coronavirus receptors

AJ Venkatakrishnan[1], Arjun Puranik[1], Akash Anand[2], David Zemmour[1], Xiang Yao[3], Xiaoying Wu[3], Ramakrishna Chilaka[2], Dariusz K Murakowski[1], Kristopher Standish[3], Bharathwaj Raghunathan[4], Tyler Wagner[1], Enrique Garcia-Rivera[1], Hugo Solomon[1], Abhinav Garg[2], Rakesh Barve[2], Anuli Anyanwu-Ofili[3], Najat Khan[3], Venky Soundararajan[1]*

[1]nference, Cambridge, United States; [2]nference Labs, Bengaluru, India; [3]Janssen pharmaceutical companies of Johnson & Johnson (J&J), Spring House, United States; [4]nference, Toronto, Canada

**Abstract** The COVID-19 pandemic demands assimilation of all biomedical knowledge to decode mechanisms of pathogenesis. Despite the recent renaissance in neural networks, a platform for the real-time synthesis of the exponentially growing biomedical literature and deep omics insights is unavailable. Here, we present the nferX platform for dynamic inference from over 45 quadrillion possible conceptual associations from unstructured text, and triangulation with insights from single-cell RNA-sequencing, bulk RNA-seq and proteomics from diverse tissue types. A hypothesis-free profiling of ACE2 suggests tongue keratinocytes, olfactory epithelial cells, airway club cells and respiratory ciliated cells as potential reservoirs of the SARS-CoV-2 receptor. We find the gut as the putative hotspot of COVID-19, where a maturation correlated transcriptional signature is shared in small intestine enterocytes among coronavirus receptors (ACE2, DPP4, ANPEP). A holistic data science platform triangulating insights from structured and unstructured data holds potential for accelerating the generation of impactful biological insights and hypotheses.

*For correspondence:
venky@nference.net

## Introduction

Since December 2019, the SARS-CoV-2 virus has been rapidly spreading across the globe. The associated disease (COVID-19) has been declared a pandemic by the WHO, with over 5 million confirmed cases and over 300,000 deaths globally as of May 23, 2020 (*Johns Hopkins Coronavirus Resource Center, 2020*). The constellation of symptoms, ranging from acute respiratory distress syndrome (ARDS) to gastrointestinal issues, is similar to that observed in the 2002 Severe Acute Respiratory Syndrome (SARS) epidemic and the 2012 Middle East respiratory syndrome (MERS) outbreak. SARS, MERS, and COVID-19 are all caused by *Coronaviruses* (CoV), deriving their name from the crown-like spike proteins protruding from the viral capsid surface. Coronavirus infection is driven by the attachment of the viral spike protein to specific human cell-surface receptors: ACE2 for SARS-CoV-2 and SARS-CoV (*Zhou et al., 2020a*; *Li et al., 2003*; *Hofmann et al., 2005*), DPP4 for MERS-CoV (*Raj et al., 2013*) and ANPEP for specific α-coronaviruses (*Yeager et al., 1992*). In addition to these receptors, the protease activity of TMPRSS2 has also been implicated in viral entry (*Hoffmann et al., 2020*; *Gierer et al., 2013*).

In a recent clinical study of COVID-19 patients from China, 48% of the 191 infected patients studied had comorbidities such as hypertension and diabetes (*Zhou et al., 2020b*). Epidemiological and

clinical investigations on COVID-19 patients have also suggested fecal viral shedding and gastrointestinal infection (*Xu et al., 2020a*; *Gu et al., 2020*; *Xiao et al., 2020*). In the case of the earlier SARS epidemic, multiple organ damage involving lung, kidney, and heart was reported (*Yang et al., 2010*). The mechanisms by which various comorbidities impact the clinical course of infections and the reasons for the observed multi-organ phenotypes are still not well understood. Thus, there is an urgent need to conduct a comprehensive pan-tissue profiling of ACE2, the putative human receptor for SARS-CoV-2.

A deep profiling of ACE2 expression in the human body demands a platform that synthesizes biomedical insights encompassing multiple scales, modalities, and pathologies described across the scientific literature and various omics silos. With the exponential growth of scientific (e.g. PubMed, preprints, grants), translational (e.g. clinicaltrials.gov), and other (e.g. patents) biomedical knowledge bases, a fundamental requirement is to recognize nuanced scientific phraseology and measure the strength of association between all possible pairs of such phrases. Such a holistic map of associations will provide insights into the knowledge harbored in the world's biomedical literature.

While unsupervised machine learning has been advanced to study the semantic relationships between word embeddings (*Mikolov et al., 2013a*; *LeCun et al., 2015*) and applied to the material science corpus (*Tshitoyan et al., 2019*), this has not been scaled-up to extract the 'global context' of conceptual associations from the entirety of publicly available unstructured biomedical text. Additionally, a principled way of accounting for the distances between phrases captured from the ever-growing scientific literature has not been comprehensively researched to quantify the strength of 'local context' between pairs of biological concepts. Given the propensity for irreproducible or erroneous scientific research (*Nature Editorial, 2016*), any local or global signals extracted from this unstructured knowledge need to be seamlessly triangulated with deep biological insights emergent from various omics data silos.

The nferX software is a cloud-based platform that enables users to dynamically query the universe of possible conceptual associations from over 100 million biomedical documents, including the COVID-19 Open Research Dataset recently announced by the White House (*The White House, 2020*; *Figure 1*). An unsupervised neural network is used to recognize and preserve complex biomedical phraseology as 300 million searchable tokens, beyond the simpler words that have generally been explored using higher dimensional word embeddings previously (*Mikolov et al., 2013a*). Our *local context* score is derived from pointwise mutual information content between pairs of these tokens and can be retrieved dynamically. Our *global context score* is derived using word2vec (*Mikolov et al., 2013a*), as the cosine similarity between 180 million word vectors projected in a 300 dimensional space (*Figure 1A*, *Figure 1—figure supplement 1*).

In order to assess the veracity of these conceptual associations derived from biomedical literature, it is absolutely essential to enable triangulation with structured data sources including gene and protein expression datasets. To address this need and empower the scientific community, we built a Single Cell RNA-seq (scRNAseq) resource (https://academia.nferx.com/) which harnesses these local and global score metrics to enable seamless integration of literature-derived associations with the analysis of transcriptomes from over 2.2 million individual cells from over 50 human and mouse tissue-types (*Figure 1B*). Here, we use this first-in-class resource to conduct a comprehensive expression profiling of ACE2 across host tissues and cell types and discuss how the observed expression patterns correlate with the pathogenicity and viral transmission shaping the ongoing COVID-19 pandemic (*Figure 1C*).

## Results

### ACE2 has higher expression levels in multiple cell types of the gastrointestinal (GI) tract compared to respiratory cells

To systematically profile the transcriptional expression of ACE2 across tissues and cell types, we triangulated scRNAseq-based measurements with literature-derived signals to automatically delineate novel, emerging, and known expression patterns (*Figure 2*; *Supplementary file 1*). This approach immediately highlights renal proximal tubular cells and small intestinal enterocytes among the cell types that most robustly express ACE2 (detection in >40% of cells). These cell types are also moderately to strongly associated with ACE2 in the literature. The strong intestinal ACE2 expression is

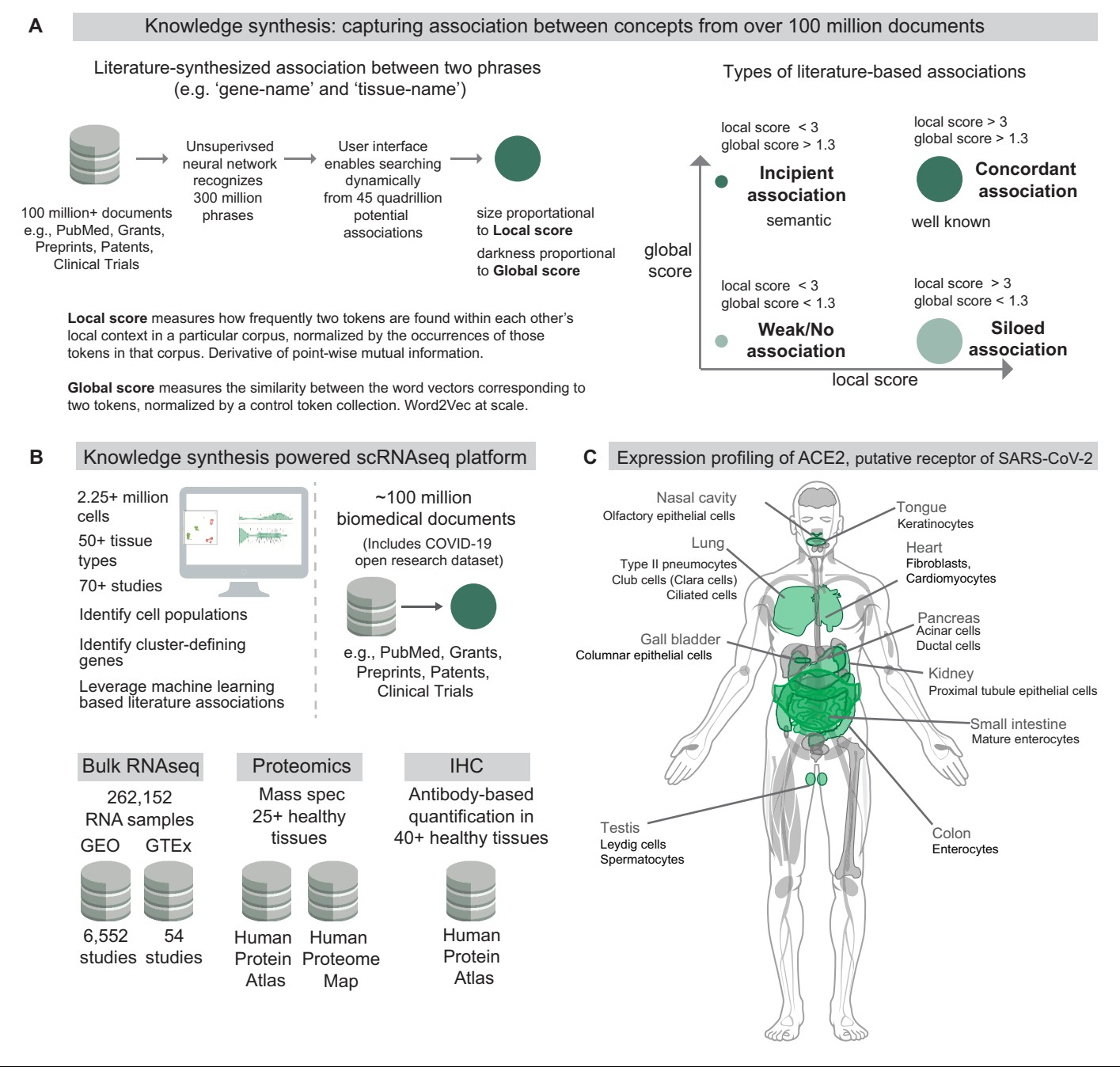

**Figure 1.** Knowledge synthesis and the nferX Single Cell resource. (**A**) Knowledge synthesis: capturing association between concepts from over 100 million documents. Schematic shows the workflow for generating literature-derived associations between phrases. Local score and global score are defined and the types of literature-derived associations are shown for combinations of high and low local and global scores. (**B**) Datasets enabling knowledge synthesis-powered scRNAseq analysis platform (https://academia.nferx.com/). Single-cell RNAseq data was obtained from publicly available human and mouse single-cell RNA-seq datasets. Bulk RNA-seq data was obtained from Gene Expression Omnibus (GEO) and the Genotype Tissue Expression (GTEx) project portal. Protein-level expression of coronavirus receptors was assessed using a collection of immunohistochemistry (IHC) images and tissue proteomics datasets from the Human Protein Atlas and the Human Proteome Map. Literature-derived association scores are obtained from over 100 million biomedical documents (**C**) Highlighting selected tissues and cell types identified by one or more modalities to express ACE2, the putative receptor of SARS-CoV-2 spike protein. Image template: https://www.proteomicsdb.org/.

The online version of this article includes the following figure supplement(s) for figure 1:

**Figure supplement 1.** Validation of metrics used to assess literature-derived associations.

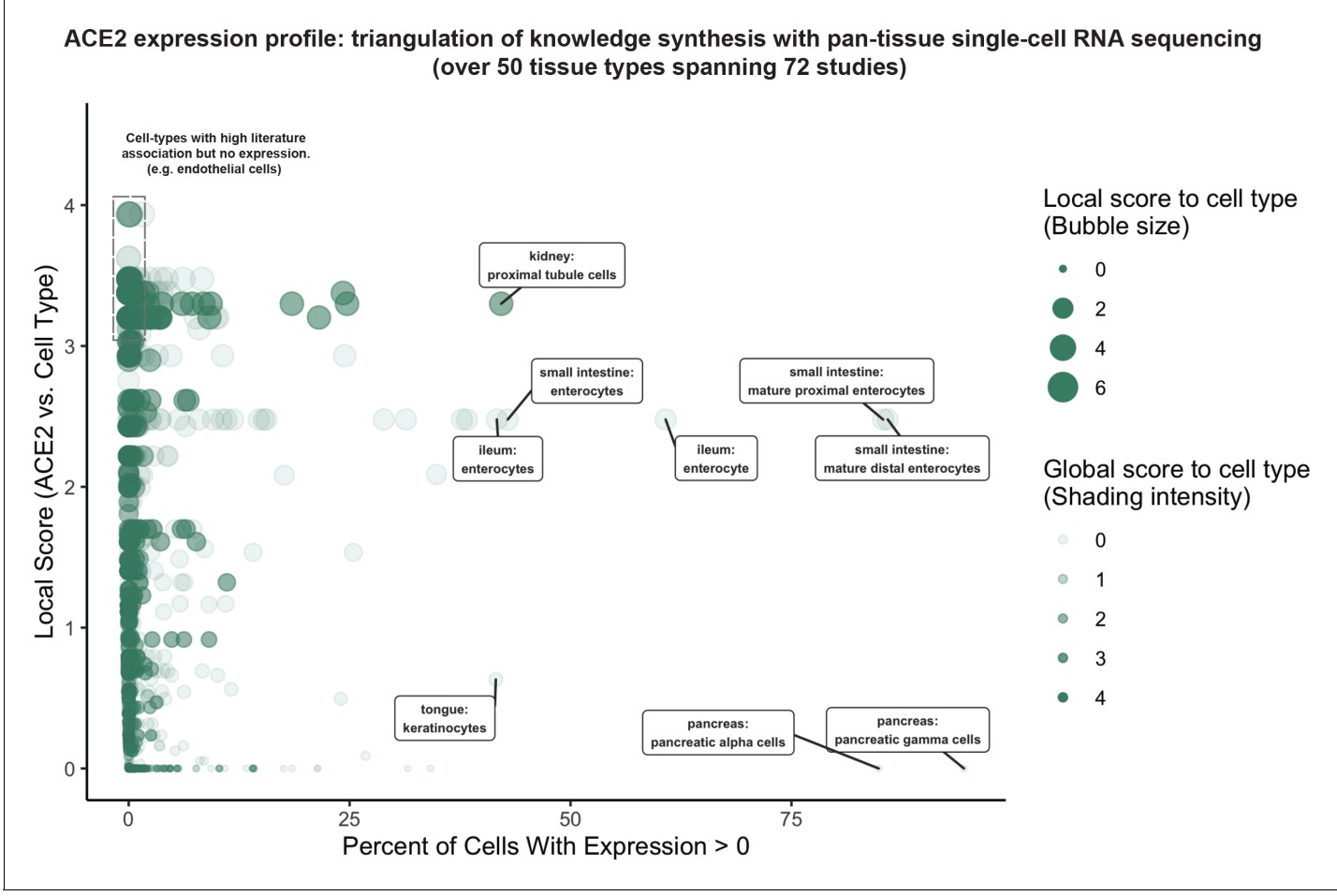

**Figure 2.** Triangulation of knowledge synthesis with ACE2 expression profile by scRNAseq across cells and tissues. Scatterplot shows comparison of percentage of cells with non-zero expression (x-axis) against literature-derived associations: local score (y-axis and size of circles) and global score (transparency of circles). Data includes cell types identified from ~1.8 million human cells and 462,000 mouse cells cumulatively from 72 studies. The online version of this article includes the following figure supplement(s) for figure 2:

**Figure supplement 1.** ACE2 expression in cell types from murine and human pancreas by scRNAseq.

**Figure supplement 2.** Multimodal analysis of ACE2 expression using bulk RNA-seq, proteomics, and IHC.

**Figure supplement 3.** ACE2, DPP4, and ANPEP show similar expression profiles in the renal proximal tubule epithelial cells by scRNAseq and bulk RNA-seq.

**Figure supplement 4.** Overview of nferX Single Cell platform functionality.

**Figure supplement 5.** Assessment of DPP4, ANPEP, and TMPRSS2 across healthy tissues using bulk RNA-seq and IHC.

**Figure supplement 6.** Single-cell RNAseq analysis of coronavirus receptors in the adult and fetal human kidney.

**Figure supplement 7.** Single-cell RNAseq analysis of coronavirus receptors in the human and murine heart.

**Figure supplement 8.** Single-cell RNAseq analysis of coronavirus receptors in adipose tissue.

**Figure supplement 9.** Single-cell RNAseq analysis of coronavirus receptors in the human and murine testis.

**Figure supplement 10.** Single-cell RNAseq analysis of coronavirus receptors in the human and murine ovary.

**Figure supplement 11.** IHC images of coronavirus receptors in healthy pancreas and liver samples from the Human Protein Atlas.

**Figure supplement 12.** Single-cell RNAseq analysis of coronavirus receptors in the human liver and pancreas.

**Figure supplement 13.** Single-cell RNAseq analysis of coronavirus receptors in human blood, spleen, and bone marrow.

**Figure supplement 14.** Single-cell RNAseq analysis of coronavirus receptors in murine spleen, bone marrow, and thymus.

**Figure supplement 15.** Single-cell RNAseq analysis of coronavirus receptors in human and murine bladder and prostate.

**Figure supplement 16.** Single-cell RNAseq analysis of coronavirus receptors in the murine uterus.

**Figure supplement 17.** Single-cell RNAseq analysis of coronavirus receptors in human and murine central nervous system tissues.

**Figure supplement 18.** IHC analysis of DPP4 expression in human brain cortex.

**Figure supplement 19.** Association of age with ACE2 expression and co-administered drugs with COVID-19 outcomes.

**Figure supplement 20.** Characterization of oral epithelium cluster-defining genes from *Xu et al., 2020b*.

**Figure supplement 21.** Expression of ACE2 in respiratory tract associated samples from GEO in comparison to Lung (GTEx).

particularly interesting given the emerging clinical reports of fecal shedding and persistence post-recovery which may reflect a fecal-oral transmission pattern (*Xu et al., 2020a*; *Gu et al., 2020*; *Xiao et al., 2020*).

Conversely, pancreatic polypeptide cells (gamma cells), pancreatic alpha cells, and keratinocytes show similarly robust ACE2 expression but have not been strongly associated with ACE2 in the literature. This combination suggests either a biological novelty or an experimental artifact. We note that the strong ACE2 expression in pancreatic cell types is derived from only one murine study (*Figure 2—figure supplement 1*; *Tabula Muris Consortium et al., 2018*), while ACE2 expression is not observed in gamma or alpha cells from scRNAseq of human pancreatic islets (*Figure 2—figure supplement 1*; *Segerstolpe et al., 2016*; *Grün et al., 2016*; *Muraro et al., 2016*). While we cannot determine the validity of either observation, this example demonstrates how knowledge synthesis can automatically surface discordant biological signals for further evaluation.

Surprisingly, cells from respiratory tissues were notably absent among the populations with highest ACE2 expression by scRNAseq (*Figure 2*). This observation was corroborated by complementary gene expression analysis of bulk RNA-seq samples from GTEx (*GTEx Portal, 2020*; *Carithers and Moore, 2015*) and the Gene Expression Omnibus (GEO) along with protein expression analysis from healthy tissue proteomics and immunohistochemistry (IHC) datasets (*Uhlén et al., 2015*; *Wang et al., 2019*; *Kim et al., 2014*), where lung and other respiratory tissues consistently show lower ACE2 expression compared to the digestive tract and kidney (*Figure 2—figure supplement 2*). However, the respiratory transmission of COVID-19 along with the disease symptomatology and well-documented viral shedding in respiratory secretions (*Wang et al., 2020c*) strongly indicates that SARS-CoV-2 indeed infects and replicates within these tissues. This would suggest that even low levels of ACE2 expression may be biologically relevant in the respiratory epithelium, and so we prioritized the respiratory and digestive tracts for further knowledge synthesis-augmented scRNAseq analysis.

We also applied the Single Cell resource to analyze several other human and mouse tissues including heart, adipose, liver, pancreas, blood, spleen, bone marrow, thymus, testis, prostate, bladder, ovary, uterus, placenta, brain, and retina. A summary of ACE2 expression across these tissues are provided in *Figure 2—figure supplements 3–18*.

## Club cells, ciliated cells, pneumocytes and nasal cavity epithelial cells are likely targets of SARS-CoV-2 in respiratory tract

Next, we classified 241 respiratory cell populations from 17 independent studies based on their expression of and literature-derived associations to ACE2 (*Figure 3A*). Consistent with the low levels of ACE2 in respiratory tissues by bulk RNA-seq, proteomics, and IHC (*Figure 2—figure supplement 2A–D*), we found that ACE2 expression is detected generally in fewer than 10% of all cell types recovered from these studies. However, as mentioned above, we believe that even low ACE2 expression levels in these respiratory cells may be relevant for COVID-19 pathogenesis.

We found that club cells (formerly known as Clara cells), were consistently among the highest-expressing respiratory cell types (*Figure 3A–B*). Literature-derived local and global scores suggest that this ACE2-club cell connection is underappreciated. We also found that ACE2 is detected in type II pneumocytes in multiple studies, although the percentage of expressing cells ranges from only 0.5–7% (*Figure 3A–B*). This relatively low expression, which may be deemed inconsequential if viewed in isolation, is strongly supported by knowledge synthesis that highlights an existing association between ACE2 and type II pneumocytes (*Figure 3A–B*). Indeed, multiple studies have demonstrated ACE2 expression in these cells (*Glowacka et al., 2011*; *van den Brand et al., 2008*; *Glowacka et al., 2010*; *Bertram et al., 2011*; *Hamming et al., 2004*; *Mossel et al., 2008*; *To and Lo, 2004*). Further, ACE2 expression in bulk-RNAseq of GTEx lung samples (n = 578) is strongly correlated to markers of type II pneumocytes, with all seven surfactant protein-encoding genes among the top 4% of transcriptional correlations to ACE2 (out of the ~19,000 genes expressed at >1 Transcripts Per Million (TPM) in GTEx lung samples; hypergeometric p-value=$1.1\times10^{-10}$) (*Figure 3—figure supplement 1*).

Our scRNAseq analysis also shows that ACE2 is expressed in small fractions of ciliated airway cells and epithelial cells of the nasal cavity (*Figure 3A–B*). While no staining is observed for ACE2 in nasopharynx samples from the Human Protein Atlas (HPA) IHC dataset (*Figure 3—figure supplement 2*), a previous IHC study did report the staining of ACE2 in nasal and oral mucosa and the nasopharynx

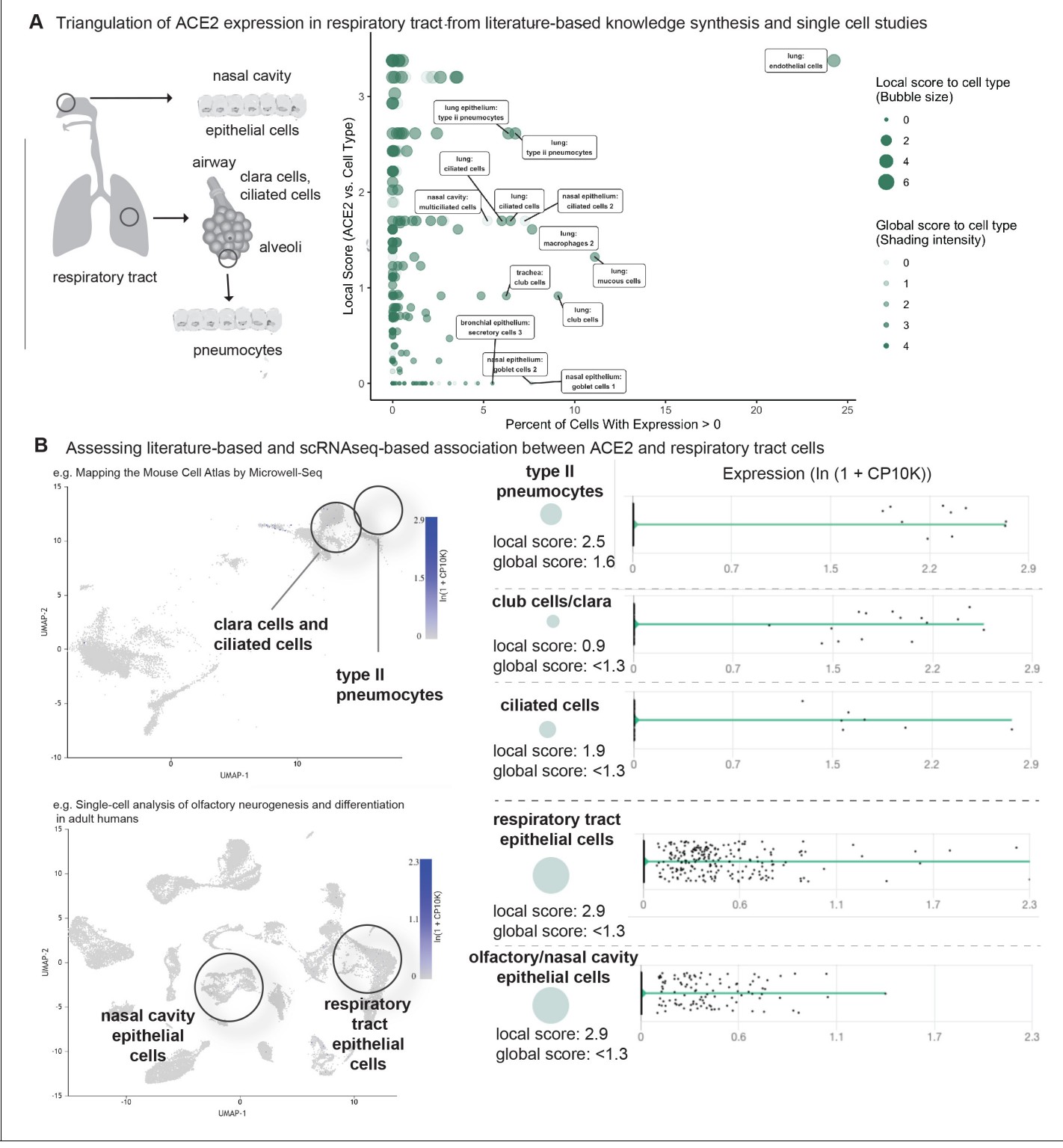

**Figure 3.** Triangulation of ACE2 expression in the respiratory tract with literature-derived insights. (**A**) Schematic representation of the respiratory system highlighting key cell types from the nasal cavity, airway and alveoli. Scatterplot shows comparison of percentage of cells with non-zero expression (x-axis) from eight single-cell studies against literature-derived associations: local score (y-axis and size of circles) and global score (transparency of circles). (**B**) Assessing literature-based and scRNAseq-based associations between ACE2 and respiratory tract cells. On the left, the dimensionality reduction plots show different cell populations associated with lung and olfactory epithelium. On the right, violin plots show the

*Figure 3 continued on next page*

*Figure 3 continued*

distribution of ACE2 expression levels in selected populations with non-zero expression. The cell types and the literature-derived local and global associations scores are shown.

The online version of this article includes the following figure supplement(s) for figure 3:

**Figure supplement 1.** ACE2 is strongly correlated to surfactant protein-encoding genes across GTEx lung samples.
**Figure supplement 2.** Negative IHC staining of ACE2 in nasopharynx from Human Protein Atlas.

(*Hamming et al., 2004*). This expression is consistent with the high SARS-CoV-2 viral loads detected in nasal swab samples (*Wang et al., 2020c*). Intriguingly, mild degeneration of olfactory epithelium was observed in an immunosuppressed animal model infected with SARS-CoV (*Schaecher et al., 2008*). These observations are correlated with emerging reports of anosmia/hyposmia (loss of smell) in otherwise asymptomatic COVID-19 patients (*ENT UK, 2020*). Such emerging clinical evidence emphasizes the need for further investigation into olfactory ACE2 expression via scRNAseq and other modalities.

Taken together, these scRNAseq analyses and triangulation to literature synthesis confirm that type II pneumocytes are a likely target of SARS-CoV-2 infection while also highlighting club cells, ciliated cells, and olfactory epithelial cells as additional potential sites of infection. Given the low ACE2 expression in the lung, it is important to identify the minimum ACE2 expression level required in a cell type for SARS-CoV-2 infection and examine whether there is correlation between the expression levels of ACE2 and propensity of the cell type to get infected.

## Tongue keratinocytes and mature small intestinal enterocytes are potential targets of SARS-CoV-2

We then classified 246 gastrointestinal cell types from 16 scRNAseq studies based on their expression of and literature associations to ACE2 (*Figure 4A*). These studies encompassed samples from the upper, mid, and lower GI tracts including tongue, esophagus, stomach, small intestine, and colon (*Tabula Muris Consortium et al., 2018*; *Han et al., 2018*; *Wang et al., 2020a*; *Smillie et al., 2019*; *Haber et al., 2017*; *HCA Data Browser, 2020*).

This analysis highlights a robust expression of ACE2 in tongue keratinocytes that has not been strongly documented in the literature, as evidenced by the weak local context score between ACE2 and keratinocytes (*Figure 4B*). In fact, we found no previous reports of ACE2 expression in keratinocytes and only one recent report suggesting ACE2 expression in the human tongue based on a combination of bulk RNA-seq and a scRNAseq dataset which has not been made publicly accessible (*Xu et al., 2020b*). We propose that a subset of ACE2$^+$ tongue keratinocytes may serve as a novel site of SARS-CoV-2 entry and highlight the need to generate additional gene and protein expression data from human tongue samples to further evaluate this hypothesis. Emerging reports of loss of taste (dysgeusia) in otherwise non-symptomatic COVID-19 patients may warrant further study of the tongue in this pathology (*Anosmia AAO-HNS, 2020*; *Shweta, 2020*).

We also found that ACE2 is highly expressed in both human and murine small intestinal enterocytes, confirming an association which has been moderately appreciated in literature, as indicated by our literature derived local score between ACE2 and enterocytes. However, to our knowledge, the transcriptional heterogeneity of ACE2 among enterocyte populations has never been explored. In this context, we found that ACE2 shows an increase in expression correlated with the maturation of murine small intestinal enterocytes, with minimal expression in stem cells and transit amplifying cells in contrast to most robust expression in mature enterocytes (*Figure 4*). To the best of our knowledge, this is the first demonstration that ACE2 expression synchronously increases over the course of enterocyte maturation. The recognition of such intra-tissue heterogeneity is necessary to specify the cell types which are most likely responsible for the proposed fecal-oral transmission of COVID-19 (*Xiao et al., 2020*).

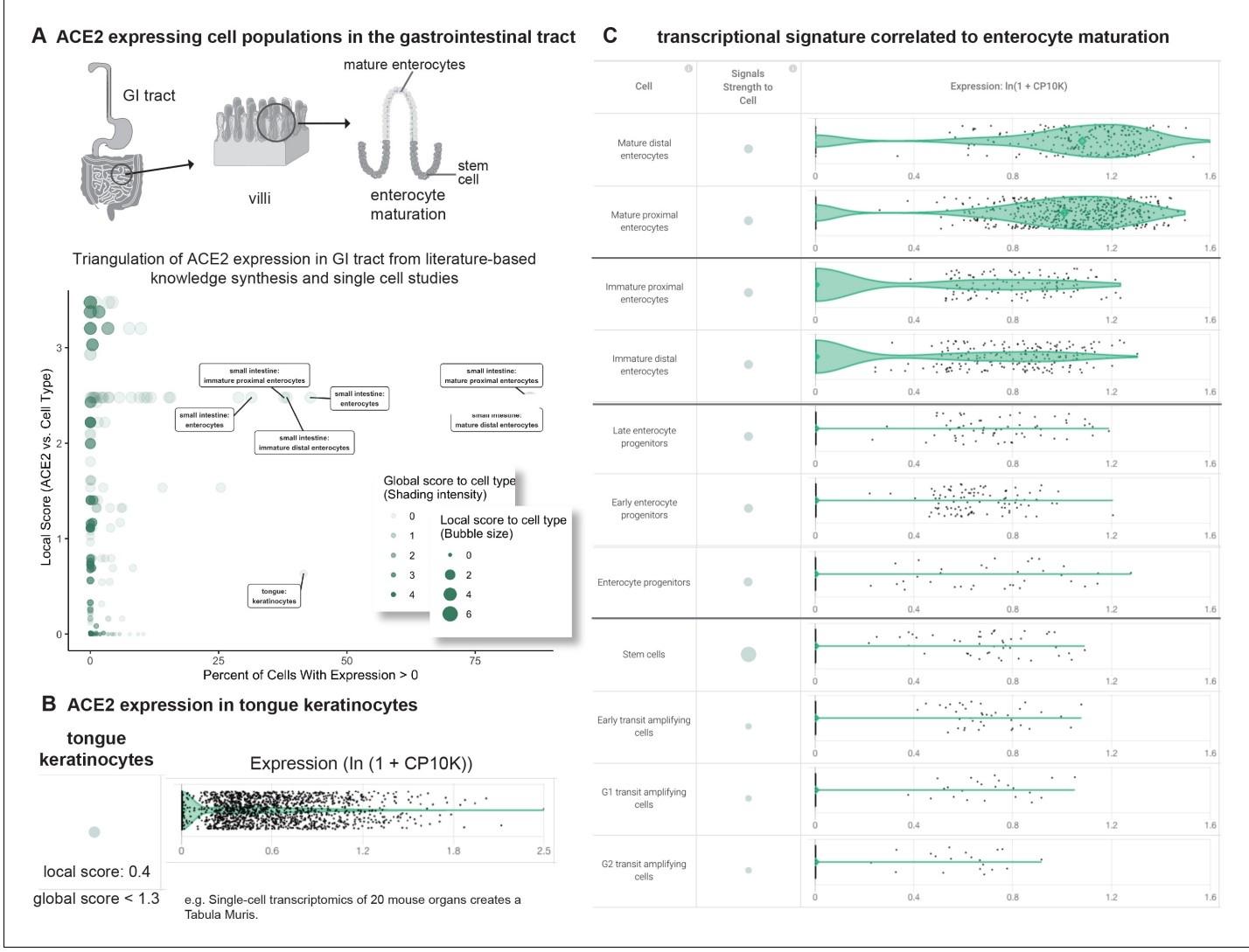

**Figure 4.** Triangulation of ACE2 expression in the gastrointestinal (GI) tract with literature-derived insights. (**A**) Schematic representation of the GI tract highlighting key cell types. Scatterplot shows comparison of percentage of cells with non-zero expression (x-axis) from nine single cell studies against literature-derived associations: local score (y-axis and size of circles) and global score (transparency of circles). (**B**) Assessing literature-based and scRNAseq-based association between ACE2 and tongue keratinocytes. Violin plot shows distributions of ACE2 expression in keratinocytes. The literature-derived local and global associations between ACE2 and keratinocytes are shown. (**C**) ACE2 transcriptional expression is correlated to enterocyte maturation. Violin plots show distribution of ACE2 expression levels in enterocytes at different stages of differentiation.

## SARS-CoV-2, SARS-CoV, MERS-CoV and HCoV-229E receptors share a transcriptional signature correlated to maturation of small intestinal enterocytes

To determine whether this maturation-correlated expression pattern is unique to ACE2, we computed cosine similarities between the ACE2 gene expression vector (Counts Per 10,000 [CP10K] values in ~6000 small intestinal enterocytes) and that of the ~15,700 other genes detected in this study (*Figure 5A*). For this analysis, the vector space is constituted of the individual cells as the dimensions using the gene expression values to construct the vectors (see Materials and methods). Interestingly, we found that ANPEP, the established entry receptor for HCoV-229E, showed the third highest cosine similarity to ACE2 (*Figure 5B*). Further, DPP4, the entry receptor for MERS coronavirus, is also among the top 1% of similarly expressed genes by this metric (*Figure 5B*). We confirmed that both of these genes do indeed show a maturation-correlated transcriptional pattern similar to that of ACE2 (*Figure 5C–D*), highlighting an unexpected shared pattern of transcriptional heterogeneity

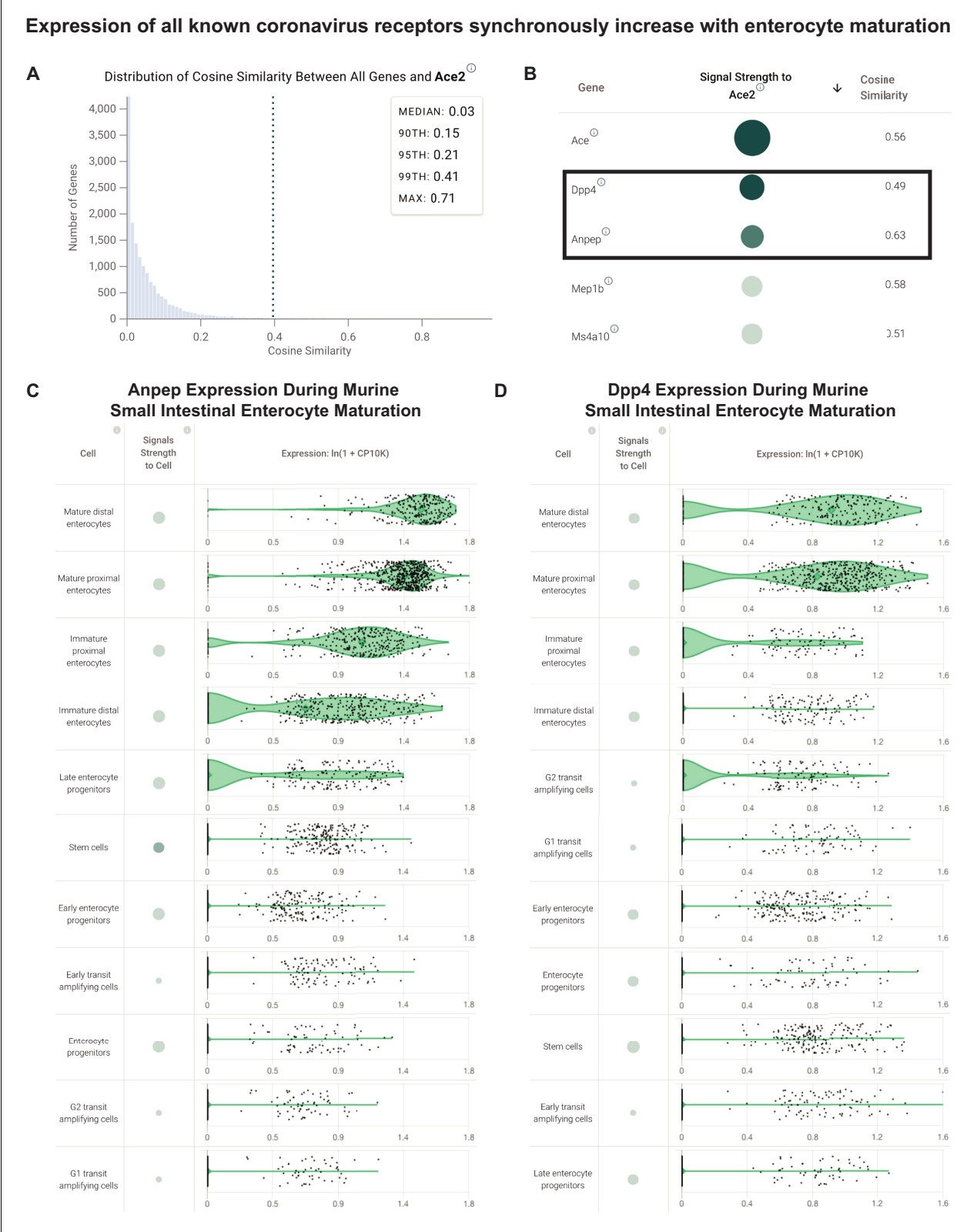

**Figure 5.** Coronavirus receptors share a transcriptional signature correlated to maturation of small intestinal enterocytes. (**A**) Distribution of cosine similarity between the 'gene expression vectors' of ACE2 and all genes in a scRNAseq study of the murine small intestine. The gene expression vector corresponds to the set of CP10K values for a given gene in each individual cell from the selected populations in the selected study. (**B**) Genes similar to ACE2 (cosine similarity >0.4) sorted by literature-derivation association. Arrow indicates a sort option available on the platform. (**C**) Transcriptional

*Figure 5 continued on next page*

*Figure 5 continued*

expression of ANPEP correlated to enterocyte maturation in murine small intestine. Violin plots show distribution of ANPEP expression levels in enterocytes at different stages of differentiation. (D) Transcriptional expression of DPP4 correlated to enterocyte maturation in murine small intestine. Violin plots show distribution of DPP4 expression levels in enterocytes at different stages of differentiation.

The online version of this article includes the following figure supplement(s) for figure 5:

**Figure supplement 1.** Coronavirus receptors show highly correlated expression patterns by single cell and bulk RNA-seq in human small intestine.

among known coronavirus receptors in a cell population which may be relevant for viral transmission.

We then asked whether this shared pattern of transcriptional heterogeneity among coronavirus receptors is observed in the human small intestine. Indeed, among all enterocytes from a human scRNAseq study, both ANPEP and DPP4 were among the top 1% of genes with similar expression vectors to that of ACE2 (*Figure 5—figure supplement 1*). We independently validated this observation by computing gene expression correlations from bulk RNA-sequencing of human small intestine samples from GTEx (n = 187), which similarly revealed that DPP4 and ANPEP are among the top 1% of correlated genes to ACE2 (*Figure 5—figure supplement 1C*). In fact, among all ~18,500 genes mean expression >1 TPM in GTEx small intestine samples, DPP4 shows the second highest correlation to ACE2 (r = 0.95).

To our knowledge, this is the first demonstration that all known coronavirus entry receptors display highly coordinated and maturation-correlated transcriptional expression patterns in intestinal epithelial cells. We propose that the requisite interaction with human proteins displaying a tightly defined expression gradient on apical surfaces of epithelial cells, which is shared among known coronavirus strains, may have fundamental implications for understanding the evolution, lifecycle, and/or transmission patterns of this family of viruses.

## Expression of coronavirus receptors in other tissues by scRNAseq

We evaluated the expression profiles of ACE2, ANPEP, DPP4, and TMPRSS2 across various other human and mouse tissues by single-cell RNA-sequencing. The complete set of studies profiled is outlined in *Supplementary file 1* and includes both tissues which were identified from bulk RNA-sequencing as potential sites of expression for one or more receptors (e.g. kidney, adipose tissue, heart, testis, prostate, blood/immune system) as well as other tissues which did not show any evidence for expression at the bulk level (e.g. brain, retina, ovary, uterus). The analysis of data from all other tissues is discussed below with corresponding images found in *Figure 2—figure supplements 3–18*.

Our analyses of bulk RNA-seq, proteomics, and IHC demonstrated that all coronavirus receptors are highly expressed in the kidney in addition to the small intestine (*Figure 2—figure supplements 2,6*). scRNAseq of human and murine kidney corroborates this finding, showing that ACE2, DPP4, and ANPEP are each robustly expressed in epithelial cells of the proximal tubules (*Figure 2—figure supplement 5*). Although clinical data suggests that SARS-CoV-2 does not reside in the urine (*Wang et al., 2020c*), we wondered whether ACE2, DPP4, and ANPEP show an overlapping pattern of expression heterogeneity among proximal tubular cells similar to that observed among small intestinal enterocytes. To address this, we computed cosine similarity scores between gene expression vectors specifically among ~27,000 recovered human proximal tubule epithelial cells (*Stewart et al., 2019*) and found that again DPP4 and ANPEP are among the genes with the most similar expression profiles to ACE2 (*Figure 2—figure supplement 3*). This is further supported by the strong transcriptional correlations among GTEx kidney samples (n = 85) by bulk RNA-seq, where DPP4 and ANPEP are again among the top 1% of gene correlations to ACE2 among the ~16,600 genes with mean expression >1 TPM (*Figure 2—figure supplement 6*). These observations are quite consistent with our previous analysis of small intestinal data and together may suggest an underlying transcriptional network which coordinates the expression of coronavirus entry receptors across diverse human tissues and cell types.

We then examined coronavirus receptor expression in the heart given the relatively high expression of ACE2 by bulk RNA-seq and the strong literature association identified between this gene-tissue pair (*Figure 2—figure supplement 7*). In the human heart, ACE2 is detected in 11% of cells or

fewer from multiple populations including smooth muscle cells, cardiomyocytes, and fibroblasts. It is detected in a higher fraction of cardiac myofibroblasts from the Tabula Muris study (*Figure 2—figure supplement 7C–D*) while showing no appreciable expression in any cardiac populations from the Mouse Cell Atlas dataset (*Figure 2—figure supplement 7E–F*). We consider this evidence inconclusive in light of the discordant IHC patterns observed in HPA (not shown). This disagreement both within and across data types highlights the heart as a tissue which certainly requires thorough follow-up regarding the intricacies of ACE2 expression.

Across multiple studies of adipose tissue, ACE2 is not strongly expressed in any cell population (*Figure 2—figure supplement 8*). Of note, this includes studies of the murine adipose stromovascular fraction (*Figure 2—figure supplement 8A–D*) along with unfractionated adipose tissue subjected to single nucleus RNA-seq (sNuc-seq) which allows for the capture and sequencing of adipocytes (*Figure 2—figure supplement 8E–F*). DPP4 and ANPEP are both detected in adipose stromal populations along with smaller fractions of immune cells. Across multiple datasets, TMPRSS2 is exclusively expressed in cells defined by canonical epithelial markers (e.g. EPCAM, KRT8, CLDN3, KRT18), suggesting epithelial contamination of the adipose tissue preparations in these studies.

In the testis, ACE2 expression was unexpectedly low by scRNAseq (*Figure 2—figure supplement 9A–D*) given its high expression by bulk RNA-seq (*Figure 2—figure supplement 2A*), strong staining by IHC, and high detection levels by proteomics (*Figure 2—figure supplement 2B–C*). The reason for this discrepancy is not clear. Based on the strong protein and bulk RNA-seq evidence, we suggest that SARS-CoV-2 may indeed be able to infect certain testicular cells, but our scRNAseq analysis did not shed light on the most likely cellular targets in this case. In both human and mouse ovary, coronavirus entry receptors and TMPRSS2 are not appreciably expressed, which is consistent with a lack of detection by our other data modalities (*Figure 2—figure supplement 10A–D*).

In the liver, expression of these genes was generally consistent with the protein expression patterns observed by IHC (*Figure 2—figure supplement 11*). ACE2 shows minimal detection throughout liver populations, while both DPP4 and ANPEP are expressed in the epithelial compartment (*Figure 2—figure supplement 12*). ANPEP expression is particularly high in the EPCAM$^+$ population of cells which may mark hepatic progenitor cells. TMPRSS2 is also expressed in these cells and in a subset of mature hepatocytes, which is discordant with the lack of TMPRSS2 staining in the liver by IHC (*Figure 2—figure supplement 12*).

In the pancreas, ACE2 is expressed in different cell types including acinar cells and ductal cells (*Figure 2—figure supplement 12*). DPP4 is robustly expressed in pancreatic alpha cells with lower expression detected in ~20% of ductal cells, while TMPRSS2 and ANPEP are both strongly expressed in the acinar and ductal populations (*Figure 2—figure supplement 12*). These patterns are largely consistent with our observations of protein expression by IHC (*Figure 2—figure supplement 11*).

To assess expression in blood and immune organs, we analyzed scRNAseq studies from blood, spleen, bone marrow, and thymus (*Figure 2—figure supplements 13–14*). Generally ACE2 and TMPRSS2 are not highly expressed in any populations from these tissues. DPP4 is expressed in subsets of T cells across these studies (*Figure 2—figure supplement 14*) along with B cells and various progenitor populations in the bone marrow from the Tabula Muris study (*Figure 2—figure supplement 14*). ANPEP expression was mostly restricted to monocytes and macrophages along with some small bone marrow progenitor populations (*Figure 2—figure supplement 14B,D*). Similarly, ANPEP expression in monocytes and macrophages may provide an alternative non-epithelial route of infection for other coronaviruses such as CoV-229E.

In both bladder and prostate samples from human and mouse, TMPRSS2 is robustly expressed in various epithelial populations (*Figure 2—figure supplement 15A–H*). The coronavirus entry receptors are only minimally detected across all bladder cell populations (*Figure 2—figure supplement 15A–F*), whereas DPP4 and ANPEP are detected in subsets of human prostate luminal cells (~8% of 18%, respectively) (*Figure 2—figure supplement 15H*). ACE2 expression is low across all recovered populations from the prostate (*Figure 2—figure supplement 15H*).

In the mouse uterus dataset of ~3700 cells from the Mouse Cell Atlas (*Figure 2—figure supplement 16A*), Ace2 was uniformly absent from all recovered populations (*Figure 2—figure supplement 16B*). Anpep is robustly detected in ~60% of glandular epithelial cells along with a smaller fraction (~10%) of stromal cells (*Figure 2—figure supplement 16B*). Dpp4 expression is detected in ~17% of dendritic cells (n = 23 of 131) along with ~5% of glandular epithelial cells, and Tmprss2 is

expressed in a similarly small fraction of the epithelial population (*Figure 2—figure supplement 16B*).

Finally, we assessed the expression of these genes in central nervous system (CNS) tissues despite their uniformly low expression and lack of detection in brain samples from GTEx and HPA, respectively (data not shown). scRNAseq data suggests similarly low expression across CNS populations, including various regions of the mouse brain and the human retina (*Figure 2—figure supplement 17*). From the Tabula Muris study, Ace2 is detected in a small number of pericytes (22 of 146) and an even lower fraction of endothelial cells (*Figure 2—figure supplement 17B*). Dpp4 is also expressed here in ~20% of endothelial cells (*Figure 2—figure supplement 17B*), which may warrant follow-up but importantly is not reflected in endothelial cells of the cerebral cortex by IHC (*Figure 2—figure supplement 18*). In all brain-derived cell populations from the Mouse Cell Atlas (*Figure 2—figure supplement 17C–D*) and human retina-derived populations (*Figure 2—figure supplement 17E–F*), these genes were uniformly not detected at significant levels.

## ACE2 expression and patient demographics

While our study profiles the expression of the coronavirus receptors from various tissue samples, whether and how much receptor expression varies among individuals across various factors like age, disease states, genetic diversity, lifestyle, and environmental factors are not well understood. For instance, in order to understand variation of expression with age we explored ACE2 expression using GTEx samples. Interestingly, ACE2 expression levels in colon (transverse) samples were higher in younger individuals in comparison to older individuals (*Figure 2—figure supplement 19A*). In contrast, ACE2 expression levels in the esophagus (gastroesophageal junction) were lower in younger individuals in comparison to older individuals (*Figure 2—figure supplement 19B*). These patterns of ACE2 expression need to be tested more rigorously on larger sample sizes across diverse ethnicities to establish statistical significance. If ACE2 were found to be expressed more significantly in the colon of younger individuals, taken together with recent reports of sustained fecal shedding of SARS-CoV-2 (*Xu et al., 2020a*; *Gu et al., 2020*), the emerging epidemiological hypothesis of younger individuals having fewer respiratory complications in general and serving as facile vectors in transmission of COVID-19 requires more deeper investigation.

The recent reports of hypertension as a comorbidity in COVID-19 patients (*Fang et al., 2020*) and specifically the use of ACE inhibitors as antihypertensives contributing to mortality encourages a hypothesis-free examination of the FDA adverse event reporting system (FAERS; see Materials and Methods). Examining the differential patterns of adverse event reports between ACE inhibitors and beta blockers - both antihypertensive drug classes, with the former known to increase ACE2 expression in select cardiovascular tissues (*Ferrario et al., 2005*) - shows that use of ACE inhibitors is associated with a higher risk of respiratory edema (*Figure 2—figure supplement 19C*). Any reports of patients on beta blockers also were removed from the ACE inhibitors set, and vice versa, for this analysis. The increased adverse events of epiglottic edema, epiglottis, pneumopericardium, upper airway obstruction, eosinophilic oesophagitis, edema mucosal, edema of the mouth, tracheal edema, palatal edema, and allergic edema associated with ACE inhibitors compared to beta blockers suggests that a thorough investigation of all 10 million plus adverse event reports is necessary, to triangulate any drug-induced side effects that also appear as comorbidities from the emerging evidence of COVID-19 mortality. Availability of single-cell RNAseq data from healthy, pathological, and drug-treated tissues would enable us to profile the age-associated and treatment-based expression levels of ACE2 across cell populations. These observations underline that investments are needed to conduct comprehensive scRNAseq profiling of tissue samples from across different demographics and pathologies pertinent to COVID-19, as such effort will hold tremendous potential to reveal under-appreciated fingerprints of coronavirus transmission patterns, tissue tropism, and mortality.

## Discussion

Recent advances in scRNAseq are empowering us to study tissue and cellular transcriptomes at previously unprecedented resolutions. Several single-cell RNA sequencing based efforts such as the Human Cell Atlas are spearheading coordinated efforts to catalog gene expression across tissues and cell types, and the raw data from many of these studies are available on public platforms such

as the Broad Institute Single Cell Portal (*Single Cell Portal, 2020a*) and Gene Expression Omnibus (GEO). Analyses of these datasets are of interest to a wide range of researchers but currently prove challenging for all but a few due to the need for specialized workflows and computing infrastructures. Consequently, the widespread use of this data for biomedical research is hampered, an issue which is particularly evident in the face of public health crises like the ongoing COVID-19 pandemic. To address this unmet need, the nferX platform Single Cell resource enables the rapid and interactive analysis of the continually growing scRNAseq datasets by specialists and non-specialists alike. Furthermore, the seamless triangulation of scRNAseq insights with global and local scores derived from the synthesis of accessible biomedical literature creates a truly first-in-class resource.

By making the resource available to all academic researchers, we enable scientists to not only dive deeper into insights that are aligned with existing knowledge but also to prioritize the novel insights which warrant further experimental validation. Looking forward, we plan to automate the integration of the rapidly growing number of scRNAseq studies so that access to the entire world's knowledge of single-cell transcriptomes is just one click away for any researcher. As we do so, we encourage interactive feedback from the scientific community so that this platform can evolve to optimally support research needs across the biomedical ecosystem, beyond the COVID-19 focus on the current study.

Combined with our analyses of bulk RNA-seq, IHC, and proteomics datasets, our characterization of the known human coronavirus receptors (ACE2, DPP4, ANPEP) using the nferX platform Single Cell resource represents the most comprehensive molecular fingerprint of host factors determining coronavirus infections including COVID-19. While this serves as a primer of the deep profiling that is made possible with this resource, we also identified several interesting aspects of coronavirus receptor biology which warrant further experimental follow-up.

We identified tongue keratinocytes and olfactory epithelia as novel ACE2-expressing cell populations and thus as important potential sites of SARS-CoV-2 infection. This molecular fingerprint is a striking correlate to established clinical reports of dysgeusia (*Anosmia AAO-HNS, 2020*) and anosmia (*ENT UK, 2020*) in COVID-19 patients, which strongly implicate the gustatory and olfactory systems in SARS-CoV-2 pathogenesis and human-to-human transmission. Tongue epithelial cells have also previously been shown to uptake Epstein-Barr virus (*Tugizov et al., 2003*), and importantly a recent study found that ACE2 is appreciably expressed in the tongue based on a small number of non-tumor bulk RNA-seq samples from TCGA (*Xu et al., 2020b*). This same study further showed by scRNAseq that ACE2 expression is observed in a subset of the human tongue (but not other oral mucosal) epithelial cells, albeit in only ~0.5% of the recovered epithelial population. This data has unfortunately not been released for public consumption but certainly does provide preliminary support for our finding, particularly as the listed set of cluster-defining genes for this population (SFN, KRT6A, KRT10) is consistent with the tongue keratinocyte identify from the *Tabula Muris* data set (*Figure 2—figure supplement 20*). We thus emphasize the imminent need for further generation of multi-omic expression data from large numbers of healthy and diseased human tongue samples drawn from a cohort of wide demographic representation.

We also observed that expression of ACE2 and other coronavirus receptors is intimately linked to the maturation status of small intestinal enterocytes, pinpointing the more mature subsets as the most likely cells to harbor SARS-CoV-2 virus. This finding amplifies the potential for fecal-oral transmission of COVID-19 (*Xu et al., 2020a*; *Gu et al., 2020*; *Xiao et al., 2020*) and should motivate further experimental validation to determine whether monitoring of fecal viral loads should be considered clinically for diagnostic or prognostic purposes.

We further found that this transcriptional mirroring of coronavirus entry receptors was not unique to the small intestine, but was also strongly present among renal proximal tubule epithelial cells, where ACE2, DPP4, and ANPEP expression tends to be observed in the same cellular subsets. These observations suggest the existence of a transcriptional network spanning tissues and cell types which may drive and regulate coronavirus receptor expression. The question of whether coronaviruses have evolved to exploit such a network may be relevant to pursue, particularly given that downregulation of ACE2 by SARS-CoV has been reported previously and is associated with poor clinical outcomes (*Glowacka et al., 2010*; *Kuba et al., 2010*). It is possible that SARS-CoV-2 is also able to induce cleavage of ACE2 as a means of viral entry and concomitant production of inflammatory cytokines (e.g. TNF-$\alpha$). However, the moderate homology of spike proteins between SARS-CoV and

SARS-CoV-2 requires additional experimentation to understand if such a mechanism operates in infection with SARS-CoV-2.

The emerging picture of the coronavirus life cycle appears to be intricately interwoven with many proteins beyond the primary host receptors. For instance, a recent structural complex of the SARS-CoV-2 spike protein with ACE2 identified SLC6A19 as an interaction partner of ACE2 (*Yan et al., 2020*). Further, spike proteins from some coronaviruses can interact with CEACAM1 (*Miura et al., 2004*) and sialylated glycans similar to influenza hemagglutinin (*Tortorici et al., 2019*) as host receptors. Future studies are likely to highlight several other proteins and glycans that constitute the 'interactome' of the coronavirus proteome. Understanding the expression profiles of the interactome across tissues will provide systems level insights on the cellular dynamics of the functional partners and the regulatory machinery of the host receptor proteins. Like in the current study, the nferX platform will be an excellent resource for unraveling the purported interaction partners for coronavirus receptors and profiling their expression across different tissues and cells constituting the human body.

Overall, this study evidences the utility of an integrative data science platform to enable rapid and high-throughput analysis of publicly available data to generate relevant biological insights and scientific hypotheses. We hope that by making our biomedical knowledge synthesis-augmented single cell platform publicly accessible, we help empower the research community to advance our understanding of the world's most pressing biomedical challenges such as COVID-19.

## Materials and methods

### Unstructured biomedical knowledge synthesis and triangulation capabilities

In order to capture biomedical literature-based associations, the nferX platform defines two scores: a 'local score' and a 'global score', as described previously (*Park et al., 2020*). Briefly, the local score is obtained from applying a traditional natural language processing technique which captures the strength of association between two concepts in a selected corpus of biomedical literature based on the frequency of their co-occurrence normalized by the frequency of each individual concept throughout the corpus. A higher local score between Concept X and Concept Y indicates that these concepts are frequently mentioned in close proximity to each other more frequently than would be expected by chance. The global score, on the other hand, is based on the neural network renaissance that has recently taken place in Natural Language Processing (NLP). To compute global scores, all tokens (e.g. words and phrases) are projected in a high-dimensional vector space of word embeddings. These vectors serve to represent the 'neighborhood' of concepts which occur around a given concept. The cosine similarity between any two vectors measures the similarity of these neighborhoods and is the basis for our global score metric, where concepts which are more similar in this vector space have a higher global score.

While the global scores in this work are computed in the embedding space of word2vec model, it can also be computed in the embedding space of any deep learning model including recent transformer-based models like BERT (*Devlin et al., 2019*). These may have complementary benefits to word2vec embeddings since the embeddings are context sensitive having different vectors for different sentence contexts. However, despite the context sensitive nature of BERT embeddings a global score computation for a phrase may still be of value given the score is computed across sentence embeddings capturing the context sensitive nature of those phrases.

From a visualization perspective, the local score and global score ('Signals') are represented in the platform using bubbles where bubble size corresponds to the local score and color intensity corresponds to the global score. This allows users to rapidly determine the strength of association between any two concepts throughout biomedical literature. We consider concepts which show both high local and global scores to be 'concordant' and have found that these typically recapitulate well-known associations.

One key aspect of the nferX platform is that it allows the user to query associated concepts for a *virtually unbounded number of possible query concepts*. This is achieved by means of two features: Firstly, the nferX platform allows the user to compose queries using the logical AND, OR and NOT operators to logically combine any number of biomedical concepts in a query, each combination

amounting to a gross or nuanced composite biomedical concept. Secondly, since logical combinations yield a virtually unbounded number of biomedical concepts that can be queries, the nferX platform implements a completely dynamic method of computing local scores on the fly by using novel high performance parallel and distributed algorithms that, in real time, scan hundreds of millions of documents to quickly locate user query related text fragments and count co-occurring biomedical concepts for computing strength of association scores and their significances.

The platform further leverages statistical inference to calculate 'enrichments' based on structured data, thus enabling real-time triangulation of signals from the unstructured biomedical knowledge graph various other structured databases (e.g. curated ontologies, RNA-sequencing datasets, human genetic associations, protein-protein interactions). This facilitates unbiased hypothesis-free learning and faster pattern recognition, and it allows users to more holistically determine the veracity of concept associations. Finally, the platform allows the user to identify and further examine the documents and textual fragments from which the knowledge synthesis signals are derived using the Documents and Signals applications.

## Association scores

Having a method that automatically consumes a corpus and computes a numeric score that captures the strength of the association between any pair of entities is obviously beneficial because then given any entity, its association strength score with all other entities can be sorted to find a sorted list of other associated entities. The number of times two entities mutually co-occur in 'small' vicinities of a corpus is the basis of all association scores. One popular traditional measure for association strength between tokens in text is pointwise mutual information, or PMI (*Evert, 2005*), which we consider in several association scores.

## Measures of association

Formally, an *association score* is some real-valued function $S(q, t)$ where $q$ is a query token/entity and $t$ is another token/entity. One important notion, the 'vicinity' of q, we formally denote as the *Context of q* : The context of q are those corpus segments deemed to be 'near' or 'local' to q. For single token queries (where q is a single entity and not a logical combination of entities) , q's context consists of all corpus segments that are 'windows' formed by taking words within a distance w (usually a tunable parameter) of words from an occurrence of q in the corpus. The dynamic adjacency engine generalizes this notion of context in a natural way to logical queries: the context for a logical q can be generalized as a certain set of fixed-length fragments.

### Co-occurrences

This is just the number of times $t$ appears in the context of $q$.

### Traditional PMI

This is $\log(p(t \mid q)/p(t))$. Here $p(t \mid q)$ is the number of times t occurs in the context of q (ie co-occurrences of t and q) divided by the total length of all q contexts in the corpus, whereas $p(t)$ is the number of occurrences of $t$ in the entire corpus, divided by the corpus length.

### Word2vec cosine similarity

The popular word2vec algorithm (*Raj et al., 2013*) generates a vector (we use 300-dimensional vector representation) for each token in a corpus. The purpose of these vectors is usually to be used as features in downstream NLP tasks. But they can also be used for similarity. The original paper validates the vectors by testing them on word similarity tasks: the association score is the cosine between the vector for $q$ and the vector for $t$. This score only applies to single-token $q$.

### Exponential mask PMI (ExpPMI)

This is our first new proposed score. PMI treats every position in a binary way: it's either in the context of $q$ or not. With a window size of say 50, a token which appears three words from a query $q$ and a token which appears 45 words from a query $q$ are treated the same. We thought it might be useful to consider a measure which distinguishes positions in the context based on the number of words away that position is from an occurrence of $q$. We did this by weighting the positions in the

context by some weight between 0 and 1. Our weighting is based on an exponential decay (which has some nice properties especially when we extend to the case of logical queries).

### Local score

This is another new proposed score. We find that PMI and ExpPMI can vary a lot for small samples (i.e. small numbers of co-occurrences, occurrences). The Local Score is *log(coocc) * sigmoid(PMI - 0.5)*, constructed to correct for this; we found that this formula too works well empirically.

### Exponential mask local score (ExpLocalScore)

We apply both modifications together: the exponential mask score is *log(weighted_coocc) * sigmoid (expPMI - 0.5)*. Here *weighted_coocc* is the sum of the weights of the positions of the corpus.

## Evaluation of literature-derived association scores

We need a notion of ground truth to evaluate the quality of association measures. We use sets of known pairs of related entities versus a 'control' group of random pairs of entities of the same classes. We use a few different sets of known pairs:

1. Disease-Gene relationships based on OMIM (*Park et al., 2020*)
2. Drug-Gene relationships (*Table 1*)
3. Drug-Disease relationships based on FDA labels
   a. Drugs and their on-label indications
   b. Drugs and their on-label adverse events
4. Logical queries for ambiguous tokens

One demonstration of the use of the logical query system is to disambiguate a token by conjoining it with a disambiguating token. An example is clearer: the token 'egfr' can refer to the gene entity *epidermal growth factor receptor*, but also the test measure entity *estimated glomerular filtration rate*. A query 'egfr AND kidney' should return results related to the latter meaning, while 'egfr AND lung_cancer' the former. In particular, an unambiguous referent to the right entity should be highly related to the query. So example known pairs in this data are ('egfr AND kidney', 'estimated_glomerular_filtration_rate') and ('egfr AND lung_cancer', 'epidermal_growth_factor_receptor'). We used an internal set of ~200–300 such ('A AND B', 'C') pairs (originally built up for other reasons).

Note: One key drawback of the word2vec vector cosine similarity (*Park et al., 2020*; *Mikolov et al., 2013b*) method is its inability to get scores for logical queries as described above,

**Table 1.** Results of evaluation.

Performance of approximately 2100 disease-gene pairs.

| Assoc score↓ | Cohen's d (+) | Mann-W U norm. (-) | Logistic log loss (-) | Logistic Brier score (-) |
|---|---|---|---|---|
| Cosine (w2v) | 1.31 | 0.197 | 0.51 | 0.168 |
| Raw PMI | 2.07 | 0.0953 | 0.374 | 0.116 |
| Raw PMI -log(pctile) | 2.15 | 0.0947 | 0.355 | 0.111 |
| Exp PMI | 2.17 | 0.0897 | 0.356 | 0.109 |
| Exp PMI -log(pctile) | 2.21 | 0.0903 | 0.341 | 0.105 |
| Raw Local Score | 2.35 | 0.0828 | 0.312 | 0.0947 |
| Raw Local Score -log(pctile) | 2.28 | 0.0832 | 0.317 | 0.0963 |
| Exp Local Score | 2.34 | 0.0812 | *0.301 | *0.0915 |
| Exp Local Score -log(pctile) | *2.36 | *0.0811 | 0.308 | 0.093 |
| log(coocc) | 2.24 | 0.097 | 0.348 | 0.105 |

Interpretation of the above table.

Each row corresponds to an association score whereas each column corresponds to one of the evaluation metrics. A (+) in the column means a higher evaluation metric value, the better the association score in that row separates the positive and random pairs. A (-) means a lower evaluation metric is better. Note all the metrics are immune to linear rescalings; also the Mann-Whitney U score is nonparametric.

because the method (*Mikolov et al., 2013b*) does not address the question of how to get vectors for queries that are logical combinations of tokens.

## Evaluation metrics

Given a scoring method and a particular set of positive/control pairs, we get two sets of scores: one set for the positive pairs and one set for the negative pairs.

Cohen's d: We compute the *Cohen's d* standard statistical measure of distance between two samples (*Cohen's D, 2016*).

Mann-Whitney U (normalized): - The Mann-Whitney U is a nonparametric measure of distribution distance: it counts the number of transposed pairs (*Contributors to Wikimedia projects, 2004*).

## Metrics based on training a 1-d logistic model

In this test, we are discriminating between two classes (true association/non-association) based on one feature. We have two metrics based on fitting a 1-feature logistic curve to the data. (*Figure 1— figure supplement 1A–B*).

Brier score: The Brier score is the average squared error of the logistic curve above: that is, for each labeled point, we square the vertical distance to the logistic curve, and average over all labeled points (*Contributors to Wikimedia projects, 2005*).

Log loss (*dansbecker, 2018*): The logistic log loss is the average *-log [model probability of true label]* for each labeled point. If the model is perfect at the point, it incurs no loss. If it predicts 0.5, it incurs *-log[0.5]* loss. If it predicts 'yes' with certainty when the answer is 'no' it incurs infinite loss (a logistic function never touches 0 or one so this won't happen in our case).

Neg log percentile: For most of the scoring rules, we also include a *-log(percentile)* version of the rule. This is constructed as follows, for query $q$, token $t$, and score $S(q, t)$:

1. Compute the scores $S(q, t')$ for $q$ with every token $t'$. Let $R$ be the number of these that are nonzero.
2. Take the rank $r$ of $S(q, t)$ among all nonzero $S(q, t')$.
3. The *neg log percentile* score $nlS(q, t)$ associated with $S$ is $-log(r/R)$

We do this to:

1. control for differences across queries
2. control for differences in the shapes of the distributions that different association scoring functions take.

This procedure maps all the $S(q, t')$ to an Exponential(1) distribution. We chose Exponential(1) because it is simple, intuitively reasonable and many of the scores naturally seemed to be approximately exponential.

## High-dimensional word embeddings for determining the significant global associations

*Figure 1—figure supplement 1C* illustrates two histograms generated from a random set of vectors (in the vector space generated by the Neural Network) where one distribution represents all vector pairs whose cosine similarity is less than 0.32 (deemed 'not strong associations') and the other distribution represents all vector pairs whose cosine similarity is greater than 0.32 (deemed 'strong associations'). This can show how common a phenomenon it is to find word vector pairs that have very good cosine similarity values but yet not co-occur even once in the corpus. The 'cosine similarity >= 0.32' bar at zero value suggests that roughly 11% of vector pairs whose cosine similarity where greater than 0.32 ('strong associations') never occurred together even once in a document. It is also clear from the figure that albeit more of the mass of the 'cosine similarity >= 0.32' distribution is skewed to the right as expected (more co-occurrences and hence unsurprisingly larger cosine similarity values), there is a long tail of the 'cosine similarity < 0.32' distribution (very high co-occurrences but small cosine similarity). The long tail is a direct consequence of negative sampling— where vectors corresponding to common words that co-occur quite often with significant words in a sliding window are moved away from vectors of the other words.

## What does the word2vec neural network do from the perspective of Genes-Diseases associations?

One way to view the word2vec 'black box' operation from a Genes/Diseases perspective (cosine of <Gene, Disease> for all Genes and Diseases) is as a Transfer Function which changed the input probability distribution (pre-training randomly assigned word vectors for Genes and Diseases) to a new probability distribution. The 'null hypothesis' (which seems to be well preserved in actuality in the way word2vec assigns random values to vectors initially) is the 'green colored' Cosine Distribution (*Figure 1—figure supplement 1D*). Once word2vec training is over, the final word vectors are placed in specific positions in the 300-dimensional space so as to present the 'blue colored' Empirical distribution (the actual cosine similarity between <Gene, Disease> pairs that we observe). The 'orange curve' is the 2-Gamma mixture (the parametric distribution that captures the 'empirical distribution' with just eight parameters (two alphas, two betas, 2 ts and two phis).

Observations from this analysis:

- Note the 'symmetrical' cosine distribution after training becomes 'Asymmetrical' with a longer 'right tail'. The asymmetry is the reason why Gamma distribution worked better than say, Gaussian, for the curve fit. The mean of the distribution gets shifted to the right after training as one would expect — the vectors during training are 'brought together' by parallelogram addition predominantly— explaining the shift to the right (negative sampling will cause a movement in the opposite direction, but that will disproportionately affect the 'ultra-high frequency' words, which get 'more' positively sampled and hence the 3-gamma with a bump near 0.6 happens for ultra-high frequency words).
- The most interesting associations, by definition, are in the tail of the distribution.

## What does varying the number of dimensions in the word2vec space do to the underlying cosine similarity distributions in a large textual corpora?

*Figure 1—figure supplement 1E* illustrates a cosine similarity probability density function (PDF) graph to visually describe the implementation of the word2vec-like Vector Space Model in various N-dimensional spaces. As described in the Materials and methods section, the system is a Semantic Bio-Knowledge Graph of nodes representing the words/phrases chosen to be represented as vectors and edge weights determined by measures of Semantic Association Strength (e.g. the cosine similarity between a pair of word embeddings represented as vectors in a large dimensional space). The cosine similarity ranges from 0 (representing no semantic association) to 1 (representing strongest association). This metric of association can reflect the contextual similarity of the entities in the Biomedical Corpora. The typical dimensionality used by our neural network for generating the Global Scores is n = 300 dimensions. This is because, as can be seen in the graph, the distribution is highly peaked with most of the mass centered around 0 – that is, a randomly chosen pair of vectors typically are orthogonal or close to orthogonal. Furthermore, over 300 dimensions, the distributions all have sufficiently long tails with the most interesting (salient) biomedical associations.

### Single-cell RNA-seq analysis platform

The objective of the single cell platform is to enable dynamic visualization and analysis of single-cell RNA-sequencing data. Currently, there are over 30 scRNAseq studies available for analysis in the Single Cell app, including studies from human donors/patients covering tissues such as adipose tissue, blood, bone marrow, colon, esophagus, liver, lung, kidney, ovary, nasal epithelium, pancreas, placenta, prostate, retina, small intestine, and spleen. Because no pan-tissue reference dataset yet exists for humans, we have manually selected individual studies to maximally cover the set of human tissues. In some cases, these studies contain cells from both healthy donors and patients affected by a specified pathology such as ulcerative colitis (colon) or asthma (lung). There are also a number of murine scRNAseq studies covering tissues including adipose tissue, airway epithelium, blood, bone marrow, brain, breast, colon, heart, kidney, liver, lung, ovary, pancreas, placenta, prostate, skeletal muscle, skin, spleen, stomach, small intestine, testis, thymus, tongue, trachea, urinary bladder, uterus, and vasculature. Note that two of these murine studies (Tabula Muris and Mouse Cell Atlas) include ~20 tissues each.

## Single-cell data processing pipeline

For each study, a counts matrix was downloaded from a public data repository such as the Gene Expression Omnibus (GEO) or the Broad Institute Single Cell Portal (*Supplementary file 1*). Note that this data has not been re-processed from the raw sequencing output, and so it is likely that alignment and quantification of gene expression was performed using different tools for different studies. In some cases, multiple complementary datasets have been generated from a single publication. In these cases, we have generated separate entries in the Single Cell platform.

While counts matrices have been generated using different technologies (e.g. Drop-Seq, 10x Genomics, etc.) and different alignment/pre-processing pipelines, all counts matrices were scaled such that each cell contains a total of 10,000 scaled counts (i.e. the sum of expression values for all genes equals 10,000 in each individual cell). All data were uniformly processed using the Seurat v3 package (*Butler et al., 2018*). In short, this pipeline involves the following steps. First, we identify 2000 variable genes across the given dataset and then perform linear dimensionality reduction by principal component analysis (PCA). Using the set of principal components which contribute >80% of variance across the dataset, we then do the following: (i) perform graph-based clustering to identify groups of cells with similar expression profiles (Louvain clustering), (ii) compute UMAP and tSNE coordinates for each individual cell (used for data visualization) and (iii) annotate cell clusters. Note that the three human pancreatic datasets (GSE81076, GSE85241, GSE86469) were integrated together in a shared multi-dimensional space using CCA (Canonical Correlation Analysis) and the integration method in the Seurat v3 package (*Butler et al., 2018*). Cell clustering and computation of dimensionality reduction coordinates were performed on this integrated dataset.

## Cell cluster annotation

In cases where publicly deposited counts matrices are accompanied by author-assigned annotations for individual cells or clusters, we have retained these cell annotations for display in the platform and accompanying analyses. For any study which was not accompanied by a metadata file containing cluster annotations, we have manually labeled clusters based on sets of canonical 'cluster-defining genes.' In these cases, we have attempted to leverage annotations and descriptions of gene expression patterns described by study authors in the manuscript text and figures corresponding to the data being analyzed.

## Metrics to summarize cluster-level gene expression

The platform allows users to query any gene in any selected study. The corresponding data is displayed in commonly employed formats including a series of violin plots and as a set of dimensionality reduction plots. Expression is summarized by listing the percent of cells expressing Gene *G* in each annotated cluster and the mean expression of Gene *G* in each cluster. To measure the specificity of Gene *G* expression to each Cluster *C*, we compute a Cohen's D value which assesses the effect size between the mean expression of Gene *G* in cluster *C* and the mean expression of Gene *G* in all other clusters. Specifically, the Cohen's D formula is given as follows: $(Mean_C - Mean_A)/(sqrt(StDev_C^2 + StDev_A^2))$ , where *C* represents the cluster of interest and *A* represents the complement of *C* (i.e. all other cell clusters). Note that this is functionally similar to the computation of paired fold change values and p-values between clusters which is frequently used to identify cluster-defining genes.

## Gene-gene cosine similarity

Within the platform, we support the run-time computation of cosine similarity (i.e. 1 - cosine distance) between the queried gene and all other genes. This provides a measure of expression similarity across cells and can be used to identify co-regulated and co-expressed genes. Specifically, to perform this computation, we construct a 'gene expression vector' for each gene *G*. This corresponds to the set of CP10K values for gene *G* in each individual cell from the selected populations in the selected study.

## Profiling expression of coronavirus receptors in single-cell datasets

For each single-cell dataset, we examined the expression of *ACE2*, *TMPRSS2*, *ANPEP*, and *DPP4*. We generally considered a cell population to potentially express a gene if at least 5% of cells from that cluster showed non-zero expression of this gene. For each dataset, we show a figure which

includes a UMAP dimensionality reduction plot colored by annotated cell type along with identical plots colored by the expression level of each coronavirus receptor in all individual cells. In some cases, we also show violin plots from the platform which automatically integrate literature-derived insights to highlight whether there exist textual associations between the queried gene and the tissue/cell types identified in the selected study.

### FDA Adverse Event Reporting System (FAERS) analysis

The FAERS application of the nferX platform supports viewing adverse event profiles of all marketed products through multiple lenses - Count, Proportional Reporting Ratio (PRR), and an nferX Adverse Event (AE) Score. $AEScore = ln(count) * 1/(1 + e^{-(prr-1.5)})$. Count is the raw number of reports between a drug and an adverse event. The proportional reporting ratio (PRR) is a simple way to get a measure of how common an adverse event for a particular drug is compared to how common the event is in the overall database. A PRR >1 for a drug-event combination indicates that a greater proportion of the reports for the drug are for the event than the proportion of events in the rest of the database, while a PRR of 2 for a drug event combination indicates that the proportion of reports for the drug-event combination is twice the proportion of the event in the overall database. The PRR is computed as follows:

$$PRR = (m/n)/((M - m)/(N - n))$$

 m = number of reports with drug and event
 n = number of reports with drug
 M = number of reports with event in database
 N = number of reports in database

Count of an event with a query drug is a good first measure of association. But it has the problem that generally common events will often show up at the top, where we are often more interested in events that are differentially associated with the query drug over other drugs. An issue with PRR is that it is noisy when the total number of event reports is small. If there are three reports of some oddly specific event and one occurs with the query drug, that event will likely have a very high PRR, but it may not be the event we would be most interested in for a drug (in FAERS such rare events are often not even proper adverse events) - we want events that occur often, and also are differentially associated with a drug - a balance between count and PRR.

The AE score tries to strike this balance in an all-in-one measure. It up-weights events that occur often for the query drug (this is the ln(count) term), and that are differentially associated with the query drug (this is the sigmoid term).

The sigmoid(PRR-1.5) term ranges smoothly from 0 to 1. It's equal to 0.5 at PRR = 1.5. When PRR = 6, sigmoid(PRR-1.5)=0.99; so PRR values >= 6 are all treated roughly equivalently by the AE score. Thus, extremely high PRRs due to small counts will not swing the AE score much beyond PRR = 6, and the ln(count) term will down-weight those small-count cases, so that they do not show up at the top of the AE score list.

A nice property of AE score is that, for a given query drug, the AE scores of the events with that drug turn out to roughly follow an exponential distribution, particularly at the tails. We can then fit exponential distributions to the scores, and analyze them. A benefit of the exponential fit is that we can make more robust claims about how significant a certain score is for a query drug, even if the empirical data is sparse/noisy at the tails for a particular drug.

## Acknowledgements

The authors thank Murali Aravamudan, Peter Lebowitz, Ajit Rajasekharan, and Mathai Mammen for their insightful reviews and feedback on our research. We also express our gratitude to Patrick Lenehan, Saurav Kumar Verma, Travis Hughes, and Vishy Thiagarajan who helped develop some of the scientific tools that were leveraged for this study.

## Additional information

### Competing interests

AJ Venkatakrishnan, Arjun Puranik, Akash Anand, David Zemmour, Ramakrishna Chilaka, Dariusz K Murakowski, Bharathwaj Raghunathan, Tyler Wagner, Enrique Garcia-Rivera, Hugo Solomon, Abhinav Garg, Rakesh Barve, Venky Soundararajan: is affiliated with nference. The author has financial interests in nference. Xiang Yao, Xiaoying Wu, Kristopher Standish, Anuli Anyanwu-Ofili, Najat Khan: is affiliated with Janssen. The author has financial interests in Janssen.

### Funding

No external funding was received for this work.

### Author contributions

AJ Venkatakrishnan, Formal analysis, Investigation, Methodology, Writing - original draft, Writing - review and editing; Arjun Puranik, Data curation, Software, Formal analysis, Validation, Investigation, Methodology; Akash Anand, Bharathwaj Raghunathan, Data curation, Software, Formal analysis, Methodology, Writing - original draft; David Zemmour, Data curation, Formal analysis, Visualization; Xiang Yao, Data curation, Formal analysis, Methodology, Writing - original draft; Xiaoying Wu, Data curation, Formal analysis, Project administration; Ramakrishna Chilaka, Data curation, Software, Visualization; Dariusz K Murakowski, Data curation, Formal analysis, Writing - original draft, Writing - review and editing; Kristopher Standish, Data curation, Formal analysis, Validation; Tyler Wagner, Enrique Garcia-Rivera, Formal analysis, Supervision, Methodology, Writing - review and editing; Hugo Solomon, Visualization; Abhinav Garg, Software; Rakesh Barve, Formal analysis, Supervision, Investigation, Methodology, Writing - original draft, Project administration, Writing - review and editing; Anuli Anyanwu-Ofili, Resources, Supervision, Project administration, Writing - review and editing; Najat Khan, Resources, Supervision, Funding acquisition, Validation, Project administration, Writing - review and editing; Venky Soundararajan, Conceptualization, Resources, Software, Supervision, Funding acquisition, Validation, Investigation, Writing - original draft, Project administration, Writing - review and editing

### Author ORCIDs

AJ Venkatakrishnan https://orcid.org/0000-0003-2819-3214
Dariusz K Murakowski https://orcid.org/0000-0002-9920-4980
Venky Soundararajan https://orcid.org/0000-0001-7434-9211

### Decision letter and Author response

Decision letter https://doi.org/10.7554/eLife.58040.sa1
Author response https://doi.org/10.7554/eLife.58040.sa2

## Additional files

### Supplementary files

• Supplementary file 1. List of studies included in the Single Cell Platform. The set of studies which have currently been analyzed and made accessible for analysis in the Single Cell Platform are listed below.

• Transparent reporting form

### Data availability

All single-cell RNAseq data used in this manuscript were obtained from published and freely available sources online. A complete list of these can be found in Supplementary file 1.

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
