## [Decision Letter]

**Acceptance summary:**

We understand that the mechanism through which SARS-CoV-2 interacts with human tissues is at best unclear. In that regard, both the nferX platform and your scientific findings are topical and are likely to be of wide-ranging interest to physicians, physician-scientists and scientists, as well as to a broader readership.

**Decision letter after peer review:**

Thank you for submitting your article "Knowledge synthesis of 100 million biomedical documents augments the deep expression profiling of coronavirus receptors" for consideration by *eLife*. Your article has been reviewed by three peer reviewers, and the evaluation has been overseen by Mone Zaidi as the Reviewing Editor and Matthias Barton as the Senior Editor. The reviewers have opted to remain anonymous.

The reviewers have discussed the reviews with one another and the Reviewing Editor has drafted this decision to help you prepare a revised submission.

As the editors have judged that your manuscript is of interest, but as described below that additional experiments and/or clarifications are required before it is published, we would like to draw your attention to changes in our revision policy that we have made in response to COVID-19 (https://elifesciences.org/articles/57162). First, because many researchers have temporarily lost access to the labs, we will give authors as much time as they need to submit revised manuscripts. We are also offering, if you choose, to post the manuscript to bioRxiv (if it is not already there) along with this decision letter and a formal designation that the manuscript is "in revision at *eLife*". Please let us know if you would like to pursue this option. (If your work is more suitable for medRxiv, you will need to post the preprint yourself, as the mechanisms for us to do so are still in development.)

This study examines single cell sequencing datasets for the expression of SARS-CoV-2 receptors in cell subsets from 25 tissues and considers potential roles in the disease. The study also connects these datasets to a novel machine learning method to additionally measure the literature associations in an unbiased manner that map back onto the single cell sequencing data. The analysis provides a deep analysis of the expression of SARS-CoV-2 receptors and allows for many interesting tissue type specific hypotheses to be entertained.

Summary:

Overall, the reviewers considered the approach to be novel and to facilitate a characterization of cells with the potential to be SARS-CoV-2 targets. The study further provides a useful resource that examines many tissues in considerable depth. Notably, the authors accurately suggest that with expanding availability of complex datasets, there is an increasing need for integrative tools that will assimilate the available information. In this respect, they use ACE2 and other putative coronavirus receptors and profile their expression across a spectrum of body tissues. The idea that this may become a broader resource for all genes was thus considered a welcome one. With that said, there are considerable issues with the analysis and integration that need to be resolved satisfactorily before further consideration.

Essential revisions:

1) The analyses seem to be confirmatory of existing literature and this approach is unable to resolve discordance between tissue expression and pathological phenotypes. As an example, while the authors demonstrate renal and intestinal hot spots of ACE2 expression with relatively lower ACE2 expression in the lungs, it is increasingly clear that the lung is the organ that drives the majority of the disease pathogenesis. Their approach is thus unable to provide novel information that could explain this discrepancy. For example, are there explorable differences between the co-expression of tissue proteases such as TMPRSS2 and ACE2 between the kidney, lung and intestines?

2) The authors describe the ambitious approach to pan-tissue profiling of ACE2 expression by applying neural network platform and triangulating with the available transcriptome/proteome data. Throughout the analysis, however, local context score and global context score did not help in identifying the correlation of ACE2 expression with the COVID-19 pathogenicity except for kidney proximal tubular cells (local context score > 3). A well-known SARS-CoV-2 reservoir, respiratory tissue, was scored insignificant which is raising the question about the platform's performance. Is there room to improve the performance of the nferX based on this study, and

do the authors think that the nferX platform is still crucial for the analysis compared to the transcriptome/proteome analysis alone other than identifying underappreciated tissue/cell types?

3) In the same context, the authors were not able to prove the synergetic performance of unsupervised machine learning from unstructured text data and the big data analysis yet trying to oversell their "deep learning" platform. They should revise the performance of their neural network platform or re-structure the manuscript without the integrating the neural network platform.

4) The authors state that hypothesis-free profiling of ACE2 expression was conducted, yet respiratory tissues were prioritized despite the low ACE2 expression and insignificant local scores (local context score < 3) derived from the neural network platform. Do the authors now consider the manual curation of clinical information necessary for the understanding of the pathogenesis?

5) Concern arises from the author's statement that even low ACE2 expression in lung cells may be sufficient for the pathogenesis. The obvious analysis would need a look at the expression in the spatial context and determine if the ACE2 expression level differs based on the location within the respiratory tissue.

6) Please discuss the findings in relation to the Cell paper (April 27).

7) The authors should consider comparing scRNAseq data and single cell proteomics data and examine the expression discrepancy. This will considerably strengthen the manuscript.

---

## [Author Response]

Summary:Overall, the reviewers considered the approach to be novel and to facilitate a characterization of cells with the potential to be SARS-CoV-2 targets. The study further provides a useful resource that examines many tissues in considerable depth. Notably, the authors accurately suggest that with expanding availability of complex datasets, there is an increasing need for integrative tools that will assimilate the available information. In this respect, they use ACE2 and other putative coronavirus receptors and profile their expression across a spectrum of body tissues. The idea that this may become a broader resource for all genes was thus considered a welcome one. With that said, there are considerable issues with the analysis and integration that need to be resolved satisfactorily before further consideration.

We thank the reviewers for their positive feedback and appreciate the opportunity to address the comments. We would like to take this opportunity to highlight that on our platform single cell RNA-seq studies are now processed continuously once made publicly available. At the time of submission, we had included 25 studies with around 1 million cells cumulatively. We have now updated the resource to include 72 studies with over 2.25 million cells cumulatively; we have also included the studies suggested by the reviewers.

Essential revisions:1) The analyses seem to be confirmatory of existing literature and this approach is unable to resolve discordance between tissue expression and pathological phenotypes. As an example, while the authors demonstrate renal and intestinal hot spots of ACE2 expression with relatively lower ACE2 expression in the lungs, it is increasingly clear that the lung is the organ that drives the majority of the disease pathogenesis. Their approach is thus unable to provide novel information that could explain this discrepancy. For example, are there explorable differences between the co-expression of tissue proteases such as TMPRSS2 and ACE2 between the kidney, lung and intestines?

We acknowledge that it may appear like there is discordance between expression level and pathological phenotypes in the case of lungs. We would like to clarify though that the minimum expression level of ACE2 required for SARS-CoV-2 infection is yet to be established. The detected expression of ACE2 in the lung is indeed relatively lower, but it is not absent. ACE2 expression is still detected notably in type II alveolar cells, club cells and clara cells of the lung and epithelial cells of the respiratory tract and nasal cavity.

Initially the lung was indeed thought to be the primary organ associated with the COVID-19 pathogenesis, but there is increasing evidence of the association of other organs such as kidney (e.g. Acute Kidney injury) and intestine (e.g. diarrhea). In Author response image 1 we show the growth of the number of COVID-19 documents associated with, for example, kidney, blood and intestine. Lung and the respiratory system may be more accessible relative to the organs and systems.

**Author response image 1. sa2fig1:** 

We agree with the reviewer that some points are confirmatory, which attests that the platform is able to recapitulate the ground truth automatically. However, we would also like to highlight that we present key findings that are completely novel. These include ACE2 expression in tongue keratinocytes and olfactory tissues, which were yet to be appreciated fully at the time of writing of the paper. Furthermore, the correlation of transcriptional signatures of coronavirus receptors with maturation stages of intestinal epithelial cells, to the best of our knowledge, is being described for the first time.In line with the reviewers’ suggestions, we have also analyzed the co-expression of TMPRSS2 and ACE2 across different cell types (Shweta FNU et al., under review at *eLife* and available on medRxiv at https://www.medrxiv.org/content/10.1101/2020.04.19.20067660v2). We observed that there is overlap of TMPRSS2 and ACE2 expression in the cell types of respiratory, intestinal and renal tissues (please see Author response image 2, Figure S1 of Shweta et al., 2020, medRxiv).

On a general note, the low detection of ACE2 in lung single cell studies may be owing to technical dropout effects. Below we cite two examples that highlight this view.From a recent article by Human Cell Atlas Lung Biological Network:

“... note that studies may lack specific cell types due to their sparsity, the challenges associated with isolation or analysis methodology. Moreover, expression may be under-detected due to technical dropout effects. Thus, while positive (presence) results are highly reliable, absence should be interpreted with care.”

Sugnak et al., Nature Medicine 2020 (nature.com/articles/s41591-020-0868-6)

“Additionally, we provide an overall note of caution when interpreting scRNAseq data for low abundance transcripts like ACE2 and TMPRSS2 because detection inefficiencies might result in an underestimation of the actual frequencies of ACE2+ or ACE2+TMPRSS2+ cells in a tissue.” Ziegler et al., Cell 2020

(https://www.sciencedirect.com/science/article/pii/S0092867420305006)

2) The authors describe the ambitious approach to pan-tissue profiling of ACE2 expression by applying neural network platform and triangulating with the available transcriptome/proteome data. Throughout the analysis, however, local context score and global context score did not help in identifying the correlation of ACE2 expression with the COVID-19 pathogenicity except for kidney proximal tubular cells (local context score > 3). A well-known SARS-CoV-2 reservoir, respiratory tissue, was scored insignificant which is raising the question about the platform's performance. Is there room to improve the performance of the nferX based on this study, anddo the authors think that the nferX platform is still crucial for the analysis compared to the transcriptome/proteome analysis alone other than identifying underappreciated tissue/cell types?

As noted by the reviewer, ACE2’s expression in the kidney proximal tubular cells was captured indeed. We note here that was done in an automated fashion. Confirming this using manual curation of a large volume of literature would be time-consuming and laborious. In general, the description of names of specific cell-types that are associated with ACE2 are still emerging in literature, which we do indeed capture with the epithelial cells of the lung, nasal cavity and respiratory tract, with local context scores tending towards 3. As more and more papers get published about COVID-19, local context scores reflect stronger association between COVID-19, ACE2 and tissues.

Furthermore, in Author response image 3 we examine the literature-based associations between ACE2 and tissue-types, and show that there is indeed significant association with respiratory tissues (lung, nasal cavity, trachea, bronchus).

**Author response image 3. sa2fig3:** 

3) In the same context, the authors were not able to prove the synergetic performance of unsupervised machine learning from unstructured text data and the big data analysis yet trying to oversell their "deep learning" platform. They should revise the performance of their neural network platform or re-structure the manuscript without the integrating the neural network platform.

We would like to clarify that we do find synergy between unstructured data and analysis of single cell RNAseq data (please see Figure 2). Please note that there are several points with significant local score and considerable ACE2 expression. Furthermore, the knowledge synthesis approach also enables us to identify the cell-types that have high-association from literature but low expression in terms of single cell expression (Figure 2). This provides a new lens to interpret the single-cell expression data For example, multiple endothelial cells have a strong association with ACE2 based on literature-derived signals, however, the single cell RNAseq expression is zero. This led us to explore endothelial cells related phenotypes and clotting has a strong literature-based association (local score: 3.5). We explored clotting systematically using lab tests of COVID-19+ patients and found it to be strongly correlated in COVID-19+ patients (plots given in Author response image 4; see also Figure 3, Pawlowski et al., 2020, medRxiv, https://www.medrxiv.org/content/10.1101/2020.05.21.20109439v1). Thus, the knowledge synthesis not only allows us to flag cell types with the knowledge from biomedical corpora, but also leads to generation of new testable hypotheses.

**Author response image 4. sa2fig4:** 

4) The authors state that hypothesis-free profiling of ACE2 expression was conducted, yet respiratory tissues were prioritized despite the low ACE2 expression and insignificant local scores (local context score < 3) derived from the neural network platform. Do the authors now consider the manual curation of clinical information necessary for the understanding of the pathogenesis?

We are certainly of the view that clinical context is relevant to understanding pathogenesis. An automated curation of biomedical document trends shows respiratory tissues to be associated with ACE2 and there are longitudinal trends in the growth of number of documents associated with COVID-19 and respiratory tissues, as highlighted above in responses to points 1 and 2. These factors motivated us to highlight the expression of ACE2 in the respiratory tissues.

5) Concern arises from the author's statement that even low ACE2 expression in lung cells may be sufficient for the pathogenesis. The obvious analysis would need a look at the expression in the spatial context and determine if the ACE2 expression level differs based on the location within the respiratory tissue.

We agree with the reviewer that the spatial context matters and would be great to analyse. At the link below, there is a list of all of the spatial datasets that are currently available using the 10x visium system. Unfortunately, there are no lung studies yet. We will be happy to update the single cell resource when lung data is available.

https://www.10xgenomics.com/resources/datasets/

6) Please discuss the findings in relation to the Cell paper (April 27).

We thank the reviewer for drawing our attention to this study. As suggested, we now discuss our findings in relation to the Cell paper: “SARS-CoV-2 Receptor ACE2 Is an Interferon-Stimulated Gene in Human Airway Epithelial Cells and Is Detected in Specific Cell Subsets across Tissues.”

Specifically, we have now analyzed data from non-human primate and human lung samples from Zeigler et al., Cell 2020. In this study, the authors identify ACE2 and TMPRSS2 co-expressing cells within lung type II pneumocytes, ileal absorptive enterocytes, and nasal goblet secretory cells. They also identify that ACE2 is an interferon-stimulated gene in vitro using airway epithelial cells. While the latter finding is distinct to this study, for the former observation in addition to these cell-types, we find a few other cell-types (e.g. kidney tubule cells) to be coexpressing ACE2 by leveraging additional publicly available datasets. We describe these in the Shweta et al. medRxiv preprint:

https://www.medrxiv.org/content/10.1101/2020.04.19.20067660v2

We would like to highlight that the data from this Cell paper is now available on the nferX single cell app. In addition, we also provide access to the single cell RNAseq data from the following SARS-CoV-2 and COVID-19 related study: “Host-viral infection maps reveal signatures of severe COVID-19 patients”. Bost et al., Cell 2020

7) The authors should consider comparing scRNA-seq data and single cell proteomics data and examine the expression discrepancy. This will considerably strengthen the manuscript.

We agree with the reviewers that comparing scRNA-seq data and single cell proteomics data would indeed help examine expression discrepancy and potentially strengthen the findings. From a practical standpoint though, single proteomics technology is still under development (please see Marx, 2019, Nature Methods, “A dream of single-cell proteomics“) compared to scRNA-seq and the data dissemination mechanisms like Gene Expression Omnibus for scRNA-seq are yet to be in place. Furthermore, since single cell proteomics is a completely different modality from scRNA-seq, building a robust software workflow is a significant undertaking. There is value in sharing findings from scRNA-seq as seen from the recent publications. Thus, in the interest of sharing this COVID-19-related resource with the scientific community in a timely manner, we prefer to release the scRNA-seq platform for now. Of course, in line with the reviewer’s suggestion, once the single cell proteomics data is available we would indeed be happy to build and share a single cell proteomics data resource as a follow up to this study.